# The Histone Methyltransferase SETD8 Regulates the Expression of Tumor Suppressor Genes via H4K20 Methylation and the p53 Signaling Pathway in Endometrial Cancer Cells

**DOI:** 10.3390/cancers14215367

**Published:** 2022-10-31

**Authors:** Asako Kukita, Kenbun Sone, Syuzo Kaneko, Eiryo Kawakami, Shinya Oki, Machiko Kojima, Miku Wada, Yusuke Toyohara, Yu Takahashi, Futaba Inoue, Saki Tanimoto, Ayumi Taguchi, Tomohiko Fukuda, Yuichiro Miyamoto, Michihiro Tanikawa, Mayuyo Mori-Uchino, Tetsushi Tsuruga, Takayuki Iriyama, Yoko Matsumoto, Kazunori Nagasaka, Osamu Wada-Hiraike, Katsutoshi Oda, Ryuji Hamamoto, Yutaka Osuga

**Affiliations:** 1Department of Obstetrics and Gynecology, Graduate School of Medicine, The University of Tokyo, Tokyo 113-8655, Japan; 2Division of Medical AI Research and Development, National Cancer Center Research Institute, Tokyo 104-0045, Japan; 3Cancer Translational Research Team, RIKEN Center for Advanced Intelligence Project, Tokyo 103-0027, Japan; 4Graduate School of Medicine, Chiba University, Chiba 263-8522, Japan; 5National Hospital Organization Tokyo Medical Center, Tokyo 152-8902, Japan; 6Tazuke Kofukai, Medical Research Institute, Kitano Hospital, Osaka 530-8480, Japan; 7Department of Obstetrics and Gynecology, Teikyo University School of Medicine, Tokyo 173-0003, Japan; 8Division of Integrated Genomics, Graduate School of Medicine, The University of Tokyo, Tokyo 113-8655, Japan

**Keywords:** chromatin immunoprecipitation sequencing, endometrial cancer, histone methyltransferases, machine learning, SETD8 TP73

## Abstract

**Simple Summary:**

The histone methyltransferase SET domain-containing protein 8 (SETD8) methylates histone H4 lysine 20 and non-histone proteins such as p53. Our aim was to determine the involvement of SETD8 in endometrial cancer and its therapeutic potential and identify the downstream genes regulated by SETD8 via H4K20 methylation and the p53 signaling pathway. We confirmed that SETD8 expression was elevated in endometrial cancer tissues. Our results suggest that the suppression of SETD8 using siRNA or a selective inhibitor attenuated cell proliferation and promoted the apoptosis of endometrial cancer cells. In these cells, SETD8 regulates genes via H4K20 methylation and the p53 signaling pathway. We also identified the prognostically important genes related to apoptosis, such as those encoding KIAA1324 and TP73, in endometrial cancer. SETD8 is an important gene for carcinogenesis and progression of endometrial cancer via H4K20 methylation.

**Abstract:**

The histone methyltransferase SET domain-containing protein 8 (SETD8), which methylates histone H4 lysine 20 (H4K20) and non-histone proteins such as p53, plays key roles in human carcinogenesis. Our aim was to determine the involvement of SETD8 in endometrial cancer and its therapeutic potential and identify the downstream genes regulated by SETD8 via H4K20 methylation and the p53 signaling pathway. We examined the expression profile of SETD8 and evaluated whether SETD8 plays a critical role in the proliferation of endometrial cancer cells using small interfering RNAs (siRNAs). We identified the prognostically important genes regulated by SETD8 via H4K20 methylation and p53 signaling using chromatin immunoprecipitation sequencing, RNA sequencing, and machine learning. We confirmed that SETD8 expression was elevated in endometrial cancer tissues. Our in vitro results suggest that the suppression of SETD8 using siRNA or a selective inhibitor attenuated cell proliferation and promoted the apoptosis of endometrial cancer cells. In these cells, SETD8 regulates genes via H4K20 methylation and the p53 signaling pathway. We also identified the prognostically important genes related to apoptosis, such as those encoding KIAA1324 and TP73, in endometrial cancer. *SETD8* is an important gene for carcinogenesis and progression of endometrial cancer via H4K20 methylation.

## 1. Introduction

Endometrial cancer is the most common cancer of the female reproductive tract, and the number of patients diagnosed with this condition has been increasing in recent years [1]. Patients with early-stage endometrial cancer have a good prognosis, but those with advanced-stage endometrial cancer have fewer chemotherapy options as only a few molecular-targeted drugs are currently approved for endometrial cancer [2]. Currently, the 5-year survival rates for endometrial cancer are 92% for stage I, 74% for stage II, 48% for stage III, and 15% for stage IV [3]. Therefore, it is necessary to understand the mechanism underlying the development and progression of endometrial cancer and to develop new molecular-targeted drugs against this disease.

Histone modification is one of the epigenetic mechanisms. More specifically, histone methylation—along with histone acetylation, phosphorylation, sumoylation, ubiquitination, and poly ADP-ribosylation—represents an important histone modification that is involved in changes in gene expression [4,5]. Histone modifications have been reported to be important in the formation of epithelial tumors; instability of histone methylation leads to cancer development and progression [6,7,8]. The expression of several histone methylases and demethylases has been reported to be upregulated in various cancers [8,9]. For instance, our previous study showed that the histone methyltransferase SUV39H2 induced chemo- and radio-resistance in lung cancer cells [10]. Additionally, the histone methyltransferase EZH2 was overexpressed in endometrial cancer cells, and knockdown of EZH2 expression as well as treatment with an EZH2-selective inhibitor resulted in cancer growth suppression and apoptosis [11]. DZNep, an agent that indirectly inhibits the activity of EZH, has also been reported as a potential therapeutic agent in cancer [12]. Therefore, the inhibition of histone methyltransferases and demethylases is a promising novel strategy for cancer therapy. However, the mechanism through which histone methyltransferase promotes carcinogenesis and cancer progression has not yet been elucidated since histone modifications regulate numerous genes by changing the 3D structure of chromatin.

Another histone methyltransferase, SETD8, methylates histone 4 lysine 20 (H4K20), which is involved in DNA damage response, mitotic condensation, and DNA replication [13,14]. The substrates of SETD8 include non-histone proteins such as p53 and PCNA [15,16,17]. Previous studies have shown that p53 methylation by SETD8 decreases tumor suppressor activity [15,16]. SETD8 is overexpressed in various types of tumors, such as bladder cancer, non-small cell lung carcinoma, and small cell lung carcinoma [17], and is associated with a shorter survival time for gastric, esophageal, and prostate cancer patients [18,19,20,21]. Accumulating evidence highlights the possibility of SETD8 being a target for anti-cancer therapeutics. For instance, UNC0379, a selective inhibitor of SETD8, improved the prognosis of neuroblastoma in preclinical xenograft models. This indicates that SETD8 is a promising therapeutic target for neuroblastoma [22,23]. We have previously reported that SETD8 could be a therapeutic target for high-grade serous ovarian cancer in gynecologic cancers [24]. However, no comprehensive analysis of SETD8 expression and function in the context of endometrial cancer has been conducted yet.

Elucidation of gene regulation via H4K20 methylation is challenging. H4K20 methylation controls gene expression activity, while others have shown that it represses transcription; a consensus has not been reached yet [25]. Chromatin immunoprecipitation sequencing (ChIP-seq) is one of the methods used to comprehensively analyze downstream genes regulated by histone modification in various types of cancers [26]. However, when numerous downstream genes are identified, it is necessary to narrow down genes that are relevant for oncogenesis. To date, a method for selecting the relevant genes from a gene cluster has not been established.

Machine learning is an artificial intelligence (AI) technology that uses specific algorithms to learn from data [27]. Recently, a few reports on big data analysis using machine learning in the field of cancer research have been published. For instance, Kanggeun et al. described an accurate cancer classification method based on mutation profiles from The Cancer Genome Atlas (TCGA) database using machine learning methods, such as random forest and deep neural network [28]. Additionally, random survival forest (RSF) is a machine-learning technique that calculates the hazard function as an ensemble of hazard functions estimated by survival trees. Similar to the random forest method, RSF is robust to outliers and can accurately assess the risk of event occurrence based on multiple factors [29].

The aim of this study was to determine the involvement of SETD8 in endometrial cancer and its therapeutic potential. We identified the downstream genes regulated by SETD8 via H4K20 methylation and p53 signal pathway using ChIP-seq and RNA sequencing (RNA-seq). From the prognostic information of TCGA, we selected the important prognostic genes with regard to endometrial cancers using RSF. Our findings provide insights into the roles of histone methylation in carcinogenesis and suggest a foundation for new effective therapeutic strategies.

## 2. Materials and Methods

### 2.1. Clinical Tissues

Freshly frozen clinical tissues of endometrial cancer (endometrioid adenocarcinoma, *n* = 49) and endometrium (*n* = 4) were obtained from the University of Tokyo Hospital (Appendix A). All patients provided written informed consent before the commencement of the study. This study was approved by the Human Genome, Gene Analysis Research Ethics Committee of the University of Tokyo (approval number: 683-19).

### 2.2. Endometrial Cancer Cell Lines

We used the HEC50B, HEC1B, ISHIKAWA, HEC151A, and HEC6 endometrioid adenocarcinoma cell lines. Cells were cultured in Eagle’s minimal essential medium (FUJIFILM Wako, Osaka, Japan; 051-07615) supplemented with 10% heat-inactivated fetal bovine serum (FBS, Thermo Fisher Scientific, Waltham, MA, USA; 10270106) and 1% penicillin/streptomycin (FUJIFILM Wako; 161-23181) and incubated at 37 °C in humidified air containing 5% CO_2_.

### 2.3. RNA Extraction, Reverse Transcription, and Quantitative Real-Time Polymerase Chain Reaction (qRT-PCR)

Freshly frozen tissues (≤30 mm^3^) were homogenized using the MagNA Lyser Instrument (Roche, Basel, Switzerland; 03358968001) and MagNA Lyser Green Beads (Roche; 03358941001). Total RNA from freshly frozen tissues and endometrioid adenocarcinoma cells was isolated using the RNeasy Mini Kit (Qiagen, Hilden, Germany; 74104), and complementary DNA was synthesized from genomic DNA-purified RNAs using qPCR-RT Master Mix with gDNA Remover (TOYOBO, Osaka, Japan; FSQ-301). The mRNA expression was measured using KOD SYBR qPCR Mix (TaKaRa Bio, Shiga, Japan; QKD-201X5) and QuantStudio 1 Real-Time PCR System (Thermo Fisher Scientific; A40425). Relative gene expression was analyzed using the 2^−ΔΔCt^ method. The primer sequences for RT-qPCR are listed in Appendix A. The experiment was performed in triplicate [24].

### 2.4. Gene Silencing

Cells were plated one day before transfection. The sequences of siRNAs specific to SETD8 (siSETD8; Sigma-Aldrich, St. Louis, MO, USA) are provided in Appendix A. MISSION^®^ siRNA Universal Negative Control #1 (Sigma-Aldrich; SIC001) was used as a control. siSETD8 (final concentration 100 nM) was transfected into cells using Lipofectamine-RNAiMAX Transfection Reagent (Invitrogen, Waltham, MA, USA; 13778150) for 2.5 h. Cells were incubated with medium containing FBS and antibiotics and used for assay [24].

### 2.5. Cell Viability Assay

Cells were transfected with siSETD8 or treated with a SETD8-selective inhibitor (UNC0379, Sigma-Aldrich; #SML-1465), cisplatin (Nichi-Iko Pharmaceutical Co., Toyama-Shi, Japan), or doxorubicin (Sigma-Aldrich) and cultured until use in the assay. Cell Count Kit-8 solution (FUJIFILM Wako; 341-07624) was added to each well and incubated for 2 h. The absorbance was measured using a microplate reader (BioTek, Winooski, VT, USA). Cells treated with dimethyl sulfoxide (DMSO, Sigma-Aldrich; D2650) were used as controls. The experiment was performed in triplicate [24].

### 2.6. Cell Cycle Analysis

Cells were fixed with 70% ethanol and incubated at 4 °C overnight. RNase A stock solution (final concentration: 0.5 mg/mL) was added and incubated for 20 min at 37 °C. Propidium iodide (PI, 50 mg/mL; Sigma-Aldrich; P4170) was added and incubated for 15 min at 4 °C in the dark. Cell cycle was measured by fluorescence-activated cell sorting (FACS) using a BD FACSCalibur™ HG Flow Cytometer Instrument (BD, Franklin Lakes, NJ, USA) and Cell Quest Pro software v3.1 (BD). FlowJo^®^ v10 (BD Biosciences, San Jose, CA, USA) was used for the analysis. The experiment was repeated three times [24].

### 2.7. Terminal Deoxynucleotidyl Transferase-Mediated dUTP Nick End-Labeling (TUNEL) Assay

siSETD8-transfected cells were cultured in Millicell EZ SLIDE 4-well glass slides (Merck, Kenilworth, NJ, USA; PEZGS0816) for 24 h and fixed with 4% paraformaldehyde/PBS solution (pH 7.4) (FUJIFILM Wako; 163-20145) for 15 min at room temperature and in 70% ethanol for 1 h at −20 °C. Fluorescein-dUTP was labeled using an in situ apoptosis detection kit (TaKaRa Bio; #MK500). Nuclei were stained with DAPI (Invitrogen; D1306) and mounted using ProLong™ Gold antifade reagent (Invitrogen; P36934). Immunofluorescence images were obtained using an LSM 700 confocal laser scanning microscope (Carl Zeiss, Jena, Germany).

### 2.8. Protein Extraction and Immunoblotting

Cells mixed with RIPA lysis buffer (FUJIFILM Wako; 188-02453) containing a protease inhibitor cocktail (Roche; 11836153001) were disrupted by ultrasonication (10 min, intermittently). Extracted proteins were boiled with 4× Laemmli sample buffer (BIO-RAD, Hercules, CA, USA; #1610747) at 95 °C for 5 min. Proteins were separated using 4–15% Mini-PROTEAN^®^ TGX™ Precast Protein Gels (BIO-RAD; #4561084) and transferred using Trans-Blot^®^ Turbo™ Mini PVDF Transfer Packs (BIO-RAD; 1704156). The primary antibodies are listed in Appendix A. Amersham ECL Select^TM^ western blotting Detection Reagent (Cytiva, Marlborough, MA, USA; RPN2235) and ImageQuant LAS 4000 mini (GE Healthcare Life Sciences, Chicago, IL, USA) were used for detection [24]. All the whole western blot figures can be found in the Appendix A.

### 2.9. Immunocytochemistry

Cells were cultured in Millicell EZ SLIDE 4-well glass slides and fixed with 4% paraformaldehyde/PBS solution (pH 7.4) (FUJIFILM Wako; 163-20145) for 1 h at 4 °C, permeabilized in 0.1% Triton^®^ X-100 (FUJIFILM Wako; 9002-93-1) for 3 min, and blocked with 3% bovine serum albumin for 1 h at room temperature. Cells were incubated with primary antibody overnight at 4 °C and secondary antibody for 1 h at room temperature (Appendix A). Immunofluorescence images were obtained using an LSM 700 confocal laser scanning microscope (Carl Zeiss) and quantified using ImageJ (U.S. National Institutes of Health, Bethesda, MD, USA).

### 2.10. Clonogenic Assay

UNC0379-treated cells were cultured in 6-well plates for 9 days. The medium was replaced with UNC0379 (0.5 and 1 μM) or 0.02% DMSO containing fresh medium every 3 days. Cells were fixed with 100% methanol for 2 h and stained with Giemsa solution (FUJIFILM Wako; 277-06995). The number of colonies (>100 cells) was counted using a stereomicroscope for the analysis. The experiment was performed in triplicate [24].

### 2.11. Survival Analysis

The mRNA expression (z-score) data of SETD8, KIAA1324, and TP73 for endometrial cancer (*n* = 526) were analyzed using the TCGA dataset from cBioPortal (https://www.cbioportal.org (accessed on 8 January 2021)). The significance was determined by the log-rank test using JMP Pro.v.16 software.

### 2.12. RNA-Seq Analysis

HEC1B and HEC50B cell lines were individually transfected with either negative control siRNA (siNC) or four SETD8-targeted siRNAs (Appendix A). After 48 h, cells were washed with ice-cold PBS (−), and total RNA was extracted using QIAzol Lysis Reagent and RNeasy Plus Mini Kit (Qiagen; 73404), according to the manufacturer’s instructions. The subsequent steps have been previously described [30].

### 2.13. ChIP-Seq Analysis

ChIP-seq experiments were conducted as previously described [26]. Antibodies (1 µg) specific for H4K20me1 (Abcam; #9051, Lot GR288167-1) were added to the sheared chromatin (10 µg) and incubated in an ultrasonic water bath for 30 min at 4 °C.

### 2.14. ChIP-qPCR

Chromatin immunoprecipitation was performed using a kit according to the manufacturer’s instructions (Cell Signaling Technology; #9003), as previously described [31], except that DNA fragments were approximately 200–500 base pairs long. Antibodies against H4K20me1 (Abcam; #9051, 1:50, Lot GR288167-1) and H4 (Cell Signaling Technology; #14149, 1:33, Lot 1) were added to the sheared chromatin and incubated for 16 h at 4 °C. Oligonucleotide primers used for ChIP-qPCR analysis are listed in Appendix A.

### 2.15. Bioinformatic Analysis

Bioinformatic analysis of RNA-seq and ChIP-seq experiments were previously described [30,31]. Gene Ontology and pathway analyses were performed using the clusterProfiler R package (v3.14.3) and visualized using Cytoscape (v3.6.1) with plug-in (ClueGOv2.5.1) [32,33,34]. We used the following ontologies: KEGG_20.11.2017 and REACTOME_Pathways_20.11.2017. To calculate enrichment/depletion tests, two-sided tests based on a hypergeometric distribution were performed. To correct for multiple testing, the Bonferroni step-down method was used. We used a min:3 max:8 GO tree with a minimum of three genes per GO term, and a kappa score of 0.4. For volcano plot representations, we used EnhancedVolcano (R package v1.4.0)

### 2.16. Machine Learning

Multivariate survival analysis using random survival forest (RSF) was performed using the random Forest SRC package v2.9.3. Here, the RSF models were built with 4000 survival trees and other default parameters. To estimate the prediction accuracy, an out-of-bag (OOB) error rate (1—Harrell’s concordance index) was used. Harrell’s concordance index is used to evaluate the prediction performance of survival analysis, which indicates how well the predicted survival time agrees with the actual measurement, with values ranging from 0 to 1; the closer the value is to 1, the better the result. For the assessment of variable importance, permutation importance was calculated by permuting OOB cases.

### 2.17. Statistical Analysis

All data and statistical analyses were performed in Excel (Microsoft Corp., Redmond, WA, USA), JMP Pro.v.16 (https://www.jmp.com/ja_jp/home.html (accessed on 8 July 2021), and R (https://www.R-project.org/ (accessed on 11 January 2022); The R Foundation, Vienna, Austria) and were used as described in Section 2.11, Section 2.12, Section 2.13, and Section 2.15 of the Materials and Methods section, except for the GraphPad Prism-generated plots (GraphPad Software, Inc., San Diego, CA, USA; v7). All image data were analyzed using ImageJ software (1.52q; U.S. National Institutes of Health). The significance values and sample sizes in the respective figures are described in the corresponding results or figure legends. Correlations were determined using the Pearson correlation coefficient. Significant differences indicated by *p*-values (* *p* < 0.01, ** *p* < 0.05) are presented in the figures and figure legends.

## 3. Results

### 3.1. SETD8 Is a Potential Therapeutic Target for Endometrial Cancer

To narrow down the histone methyltransferase that is a potential therapeutic target for endometrial cancer, we first performed qRT-PCR to determine the expression of several histone methyltransferase genes in endometrial cancer tissues (data not shown). *SETD8* tended to be overexpressed in endometrial cancer tissues (*n* = 49) compared with that in normal endometrial tissues (*n* = 4) (Figure 1A). Next, to determine whether SETD8 plays a critical role in the proliferation of endometrial cancer cells, we performed knockdown experiments using small interfering RNAs (siRNAs) specific to SETD8 (siSETD8 #1 and #2) in endometrial cancer cells. The siSETD8-transfected cells showed significant growth suppression (Figure 1B). In addition, they showed decreased mono-methylation levels of H4K20me1 and increased PARP cleavage, a hallmark of apoptosis (Figure 1C). To confirm whether SETD8 knockdown was related to cell cycle and apoptosis, we performed flow cytometry and TUNEL assays and observed a significantly increased proportion of cells in the subG1 phase, whereas the proportion of cells in the G1 phase was significantly decreased. Furthermore, TUNEL-positive cells were detected in siSETD8-transfected cells (Figure 1D,E). These results indicate that SETD8 knockdown induces cell cycle arrest and apoptosis in endometrial cancer cells. Treatment with the SETD8-selective inhibitor UNC0379 suppressed cell proliferation concomitant with reduced H4K20me1 levels in a dose-dependent manner (Figure 2A,B), while the half-maximal inhibitory concentration (IC50) of UNC0379 in endometrial cancer cells ranged from 576 to 2540 nM (Figure 2A). UNC0379-treated cells induced PARP cleavage (Figure 2B), significantly increased the proportion of cells in the sub-G1 phase (Figure 2C), and decreased the proportion of cells in the G1 phase. These results also indicated that inhibition of SETD8 induced cell cycle arrest and apoptosis in endometrial cancer cells. Furthermore, the clonogenic assay showed a decrease in the number of colonies in a dose-dependent manner (Figure 2D). Combination treatment with UNC0379 and doxorubicin or cisplatin additively inhibited cell proliferation (Figure 2E).

### 3.2. SETD8 Regulates Expression of Multiple Genes That Are Highly Associated with the P53 Pathway

To identify the downstream genes regulated by SETD8, we performed RNA-seq in the endometrial cancer cell lines, HEC50B and HEC1B, transfected with negative control siRNA (siNC) and SETD8-targeting siRNAs (siSETD8), respectively. Since there was slight variation between each replicate and the siSETD8-treated group was well separated from the negative control group, siRNA’s off-target was considered low (Figure 3A). Integrative analysis of gene expression profiles indicated that the expression levels of 1551 genes were upregulated by attenuating SETD8 expression in HEC50B cells. In contrast, the expression of 335 genes was downregulated (Figure 3B, Appendix A). In HEC1B cells, the expression levels of 1633 genes were upregulated and those of 250 genes were downregulated (Appendix A, Appendix A). Pathway enrichment analysis showed that SETD8 regulates several pathways, including the p53, MAPK, and PI3K-Akt signaling pathways. Notably, we found that these pathways are not only significantly enriched but also systematically connected (Figure 3C,E). Additionally, using MCODE, a cytoscape plug-in for discovering protein interaction networks, we found that genes whose expression increased following SETD8 knockdown were involved in cell proliferation, angiogenesis, and infection, and these genes interacted with p53 (Figure 3D and Appendix A). Gene Ontology analysis revealed that the molecular transducer activity and signaling receptor activity were enriched in SETD8-knockdown endometrial cancer cells (Figure 3F), consistent with the systematic interconnection among the pathways we identified above (Figure 3C,E). Next, we examined individual gene expression that fluctuated significantly among the p53-related pathway following SETD8 knockdown in HEC1B (Figure 4A) or HEC50B (Figure 4B). In addition, expression of 11 (*TP73*, *SFN*, *GADD45A*, *CDKN1A*, *PERP*, *CYCS*, *SERPINE1*, *GADD45G*, *CD82*, *PMAIP1*, and *GADD45B*) of the 74 genes related to the KEGG p53 pathway was commonly upregulated (Figure 4C). Of these 11 genes, four genes (*SFN*, *CDKN1A*, *GADD45G*, and *TP73*) were identified in the TCGA database for endometrial cancer, which showed a positive correlation between expression and prognosis (Figure 4D).

### 3.3. SETD8 Regulates KIAA1324 and TP73 Expression via H4K20 Methylation

To identify target genes regulated by SETD8 via H4K20 methylation, we performed H4K20 methylation ChIP-seq in HEC50B cells transfected with siNC or siSETD8. The ChIP-seq data indicate that H4K20 mono-methylation was enriched in intragenic regions (Figure 5A), consistent with previous results [22]. We identified 72 genes regulated by SETD8 via H4K20 methylation (Figure 5B). Expression of almost all these genes was upregulated by attenuating H4K20 methylation following SETD8 knockdown in endometrial cancer cells, indicating that H4K20 methylation mainly suppresses gene expression (Figure 5B). To identify the important genes among the 72 genes whose expression levels are regulated by SETD8 via H4K20 methylation, we used the random survival forest (RSF) method. RSF is a machine learning method for non-parametric multivariate survival analysis, which estimates the non-linear effects of multiple variables on prognosis and is also capable of selecting the relevant variables for the prediction of clinical prognostic data (OS: overall survival, DFS: disease-free survival) of endometrial cancer from the TCGA database used for RSF analysis (Figure 5C). Although the main purpose of RSF analysis was to identify important genes associated with prognosis, the Harrel’s C-index values for OOB data were 0.62 and 0.48 for OS and DFS, respectively. Nevertheless, we identified several important prognostic genes for endometrial cancer using random forest analysis (Figure 5C). Among these genes, we selected *KIAA1324* and *TP73*, which are involved in apoptosis and highly important for OS and DFS prognosis predicted by RSF (Figure 5D). H4K20me1 ChIP-qPCR on SETD8-knockdown endometrial cancer cells further confirmed that the expression of *KIAA1324* and *TP73* was indeed regulated by SETD8 via H4K20 methylation (Figure 5E).

Survival analyses showed a strong correlation between OS prognosis and *TP73* as well as *KIAA1324* expression (Figure 6A). To confirm whether SETD8 regulates *KIAA1324* and *TP73*, we performed quantitative real-time qPCR, immunoblotting, and immunocytochemistry with siSETD8-transfected and UNC0379-treated endometrial cancer cells. Expression levels of *TP73* and *KIAA1324* were significantly upregulated in siSETD8-transfected and UNC0379-treated endometrial cancer cells (Figure 6B–D).

## 4. Discussion

We confirmed that SETD8 expression was elevated in endometrial cancer tissues compared with that observed in normal endometrial tissue. Our in vitro results from several endometrial cancer cell lines further suggest that the suppression of SETD8 using siRNA or a selective inhibitor attenuated cell proliferation and promoted apoptosis. SETD8 regulates genes via H4K20 methylation and p53 signaling pathway in endometrial cancer cells. Using machine learning, we identified the important prognostic genes related to apoptosis, such as *KIAA1324* and *TP73*, from a large set of genes influenced by SETD8. Overall, these data indicate the potential of SETD8 as a new therapeutic target for endometrial cancer. SETD8 is overexpressed in various types of cancers, including lung, gastric, and renal cancers [17,18,35]. Regarding gynecological cancer, we analyzed the expression of SETD8 in high-grade serous ovarian cancer (HGSOC) cells [24]. It is considered a potential novel therapeutic target for this gynecological tumor. RT-PCR showed a high expression of SETD8 in HGSOC cells. Consistent with these previous reports, our RT-PCR data showed that SETD8 tended to be overexpressed in endometrial cancer tissues. Similar to that in other previously reported carcinomas, knockdown of SETD8 or the addition of selective inhibitors significantly suppressed cell growth in endometrial cancer cell lines [24]. Thus, we propose that inhibition or knockdown of SETD8 suppresses the proliferation of endometrial cancer cells by regulating the G1/S cell cycle and promoting apoptosis. Our results indicate that G1/S arrest occurred early after SETD knockdown, followed by the induction of apoptosis in endometrial cancer cell lines. Consistent with these results, several research groups have proposed that SETD8 plays a role in controlling G1/S [16,36].

Although findings about the anti-tumor effects of SETD8 inhibitors in several types of cancers have been previously reported, this is the first report on endometrial cancer [19,20,21,22,23,24,37,38]. Moreover, SETD8 inhibitors showed long-term effects against endometrial cancer in the clonogenic assay. Doxorubicin and cisplatin are key drugs used for the treatment of endometrial cancer [39,40]. Our results suggest that a combinatorial therapy composed of a small-molecule SETD8 inhibitor such as UNC0379 and conventional chemotherapy could represent a plausible strategy for a more effective treatment of endometrial cancer.

RNA-seq data indicate that the expression levels of 1887 genes were significantly altered by attenuating SETD8 expression in endometrial cancer cell lines. Furthermore, pathway analysis showed that genes significantly altered by SETD8 are involved in several pathways such as the p53 signaling pathway, MAPK signaling pathway, and uptake by IGFBPs. IGFBPs and related proteins are associated with endometrial cancer and may represent a risk factor, supporting our data [41,42]. Additionally, gene ontology analysis indicated that the genes involved in cellular proliferation, angiogenesis, and infection were highly interconnected with p53 in SETD8 knockdown in endometrial cell lines.

The signaling pathways associated with phospholipase C, ILK1/integrin, CXCR4, Rho GTPase signaling, and actin nucleation are altered by SETD8 depletion in medulloblastoma [43]. Compared with this previous report, there was no concordant pathway in the significantly altered pathways of endometrial cancer cell lines after SETD8 knockdown. Thus, it is speculated that the pathway of action regulated by SETD8 varies depending on the type of tumor. SETD8 suppresses p53 activity via methylation [15,16,23]. Thus, our results indicate that SETD8 regulates multiple genes and pathways via p53 methylation. In addition, we extracted p53-related genes that were significantly upregulated in the two endometrial cancer cell lines (HEC1B and HEC50B) in which SETD8 was knocked down. Among them, we identified four genes (*SFN*, *CDKN1A*, *GADD45G*, and *TP73*) whose expression was positively correlated with endometrial cancer from TCGA data. SFN is an anti-cancer substance found in broccoli, which inhibits cell growth and induces apoptosis in various cancer cells [44]. Expression of CDKN1A, also known as p21, is regulated independently of p53 and is known to regulate the cell cycle by inhibiting cyclin-dependent kinases and inducing apoptosis [45]. Only one study reported that growth arrest and DNA damage-inducible gamma (GADD45G) exerts anti-tumor effects; however, no detailed investigation has been conducted in cancers so far [46]. By inhibiting p53 methylation via SETD8 knockdown, the expression of these genes may be enhanced, leading to apoptosis and cell cycle arrest.

Using ChIP-seq, we identified 42 genes regulated via H4K20me methylation. Previous studies have shown that SETD8 plays a key role in the proper double-strand break response, changes in higher-order chromatin, and regulation of proper DNA replication [25]. As noted above, reports on gene regulation via H4K20 methylation have been inconclusive. For example, previous studies reported that H4K20me1 is enriched mainly downstream of sites where transcription is active [25,47]. Furthermore, the level of H4K20me1 in gene bodies was positively correlated with the level of gene expression during cell differentiation [48]. Conversely, a SETD8 loss-of-function assay showed that H4K20me1 represses the transcription of target genes [49,50,51]. Therefore, it remains unclear whether H4K20 methylation promotes or inhibits transcription. Our ChIP-seq and RNA-seq data allowed us to identify 72 genes regulated via H4K20 methylation in an endometrial cancer cell line. Most of the 72 genes were upregulated by the knockdown of SETD8, thus suggesting that H4K20 methylation mainly promotes transcriptional repression.

To confirm the genes regulated by H4K20 methylation, we must perform individual ChIP-PCR for each gene. However, due to the multi-step nature of ChIP-qPCR, performing ChIP-qPCR for 42 genes is challenging. To date, there has been no standard way to narrow down the most relevant genes for a certain process when handling data from several genes. To efficiently select the relevant genes, we used a random forest classification, which is a machine-learning method initially proposed by Leo Breiman in 2001 and used for classification, regression, and clustering [52]. Since there are non-linear effects for genes and interactions between genes, the conventional Cox regression analysis cannot properly assess the effect [52]. This is an amenable learning algorithm that uses a decision tree as a weak learner and is suitable for examining which parameters are important [52]. We examined the expression of the 72 genes and their importance for OS and DFS using endometrial cancer patient data from the TCGA database and the random forest method. *KIAA1324* and *TP73* were two of the top five genes related to apoptosis, suggesting their prognostic importance for endometrial cancer prognosis. The *KIAA1324* gene, also known as EIG121 (estrogen inducible gene 121), encodes a 1013-amino acid transmembrane protein that is highly conserved among organisms and shown to induce apoptosis by suppressing cancer protein GRP78 [53]. A positive correlation between *KIAA1324* expression and endometrioid as well as high-grade serous ovarian cancer prognosis has been previously reported [54]. *KIAA1324* expression has been reported to decrease with the progression of endometrial cancer [55]. Consistent with previous reports, TCGA data suggested that *KIAA1324* expression is positively correlated with prognosis. Furthermore, *KIAA1324* expression is reportedly low in endometrial cancer; however, the reason underlying the low expression remains unknown. Our analysis indicates that SETD8 overexpression in endometrial cancer regulates the expression of *KIAA1324* via histone methylation, which might explain the previously identified low *KIAA1324* expression levels.

TP73 is a member of the p53 family and is structurally and functionally similar to p53 [56]. TP73 is involved in apoptosis, genomic stability, and autophagy [57]. Epigenetic analyses have indicated that TP73 is activated by DNA methylation in the promoter regions [58]. However, there are no previous reports indicating that histone methylation regulates TP73.

Here, we report for the first time that histone methylation by SETD8 regulates TP73 expression. In contrast to *TP53*, which is most commonly mutated in cancer, TP73 is rarely mutated despite its role in tumor suppression [59]. Since the low expression of *TP73* in cancer is regulated by DNA methylation and histone modification, *TP73* is considered a good therapeutic target for cancer. In contrast to some reports associated with *KIAA1324* expression, there are no reports on the correlation between *TP73* expression and prognosis of endometrial cancer patients. Therefore, we conducted a detailed investigation of the relationship between *SETD8* and *TP73* expression levels. Our immunoblotting and immunocytochemistry data suggest that SETD8 knockdown with siRNA or a SETD8-selective inhibitor in endometrial cancer cell lines led to a decrease in H4K20 methylation, an increase in *TP73* expression, and ultimately, apoptosis. These results suggest a mechanism through which SETD8 overexpression in endometrial cancer promotes the development of cancer by suppressing the expression of *TP73* via H4K20 methylation and p53 methylation, thus inhibiting apoptosis (Figure 6E). This reveals a novel mechanism underlying the anti-tumor effect of SETD8 inhibition in endometrial cancer. Nevertheless, the present study has several limitations. First, we did not perform in vivo experiments to ascertain whether SETD8 inhibitors represent potential therapeutics in endometrial cancer. Second, although we identified new prognostic genes such as *KIAA1324* and *TP73*, it is possible that other genes regulated by H4K20 methylation are involved in endometrial cancer development. Third, in this case, machine learning was used only as an experimental method to select important genes. The purpose of using this method was not to create a risk prediction model. However, the risk assessment itself using machine learning is useful, and we plan to work on this aspect in future research. Additionally, we need to consider the limitation of the RSF model carefully because RSF enables highly accurate predictions by using information on a large number of gene expressions. It will be necessary to develop a method to make predictions based on as little gene expression information as possible when considering clinical applications in the future.

Our present findings highlight the role of SETD8 overexpression in endometrial cancer, similar to other cancer types, suggesting that it might represent a novel therapeutic target. Mono-chemotherapy with selective SETD8 inhibitors such as UNC0379, or a combination of chemotherapy using a selective SETD8 inhibitor and conventional anti-cancer drugs might be a promising strategy to improve the outcome of patients with high-risk endometrial cancer. Additionally, in endometrial cancer, SETD8 was found to function indirectly or act through a pathway that represses the function of a tumor suppressor gene via H4K20 methylation and p53 expression. In particular, with respect to the previously poorly understood gene regulatory mechanism of H4K20 methylation, our results suggest that H4K20 methylation is involved in transcriptional repression.

## 5. Conclusions

In conclusion, *SETD8* and *TP73* are important genes for the carcinogenesis and progression of endometrial cancer and represent new therapeutic targets that need to be evaluated in the future.

## Figures and Tables

**Figure 1 cancers-14-05367-f001:**
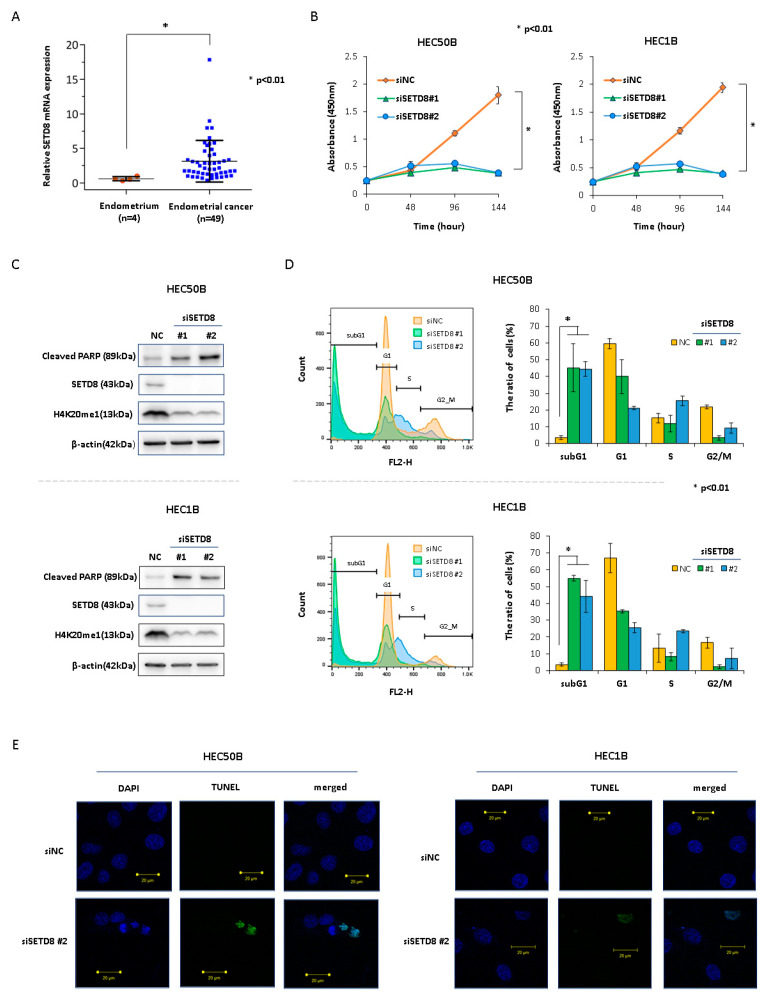
SETD8 expression is elevated in endometrial cancer, and SETD8 knockdown suppresses cell proliferation and induces apoptosis in endometrial cancer cells. (**A**) *SETD8* mRNA expression was increased in endometrial cancer tissues as analyzed using qRT-PCR (*p* < 0.01). (**B**) Transfection of siSETD8 suppressed cell proliferation. (**C**) HEC50B and HEC1B cells transfected with siSETD8 for 48 h. siSETD8-transfected cells showed decreased H4K20me1 levels and increased cleaved PARP levels. (**D**) HEC50B and HEC1B transfected with siSETD8 for 96 h. SiSETD8-transfected cells showed prolonged sub-G1 phase and cell cycle arrest. (**E**) TUNEL-positive cells were detected in siSETD8-transfected cells (HEC50B and HEC1B cells).

**Figure 2 cancers-14-05367-f002:**
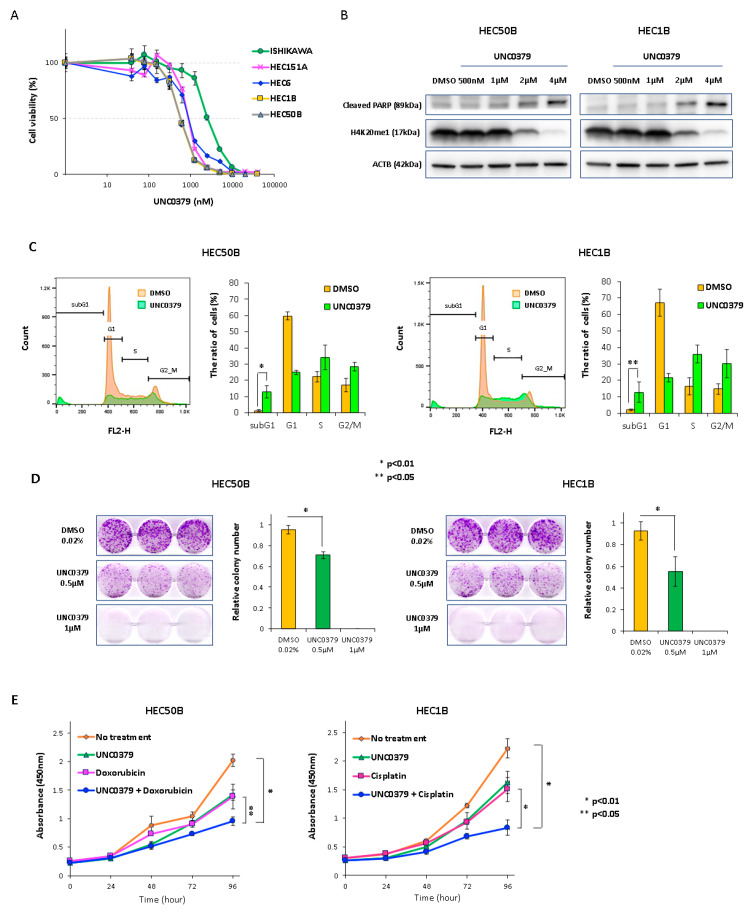
Effects of UNC0379 on endometrial cancer cells. (**A**) Endometrial cancer cells were treated with a SETD8-selective inhibitor (UNC0379) for 4 days. The IC50 of UNC0379 ranged between 576 nM (HEC50B) and 2540 nM (ISHIKAWA). (**B**) HEC50B and HEC1B cells were treated with various concentrations of UNC0379 for 96 h. UNC0379-treated cells showed decreased H4K20me1 levels and increased cleaved PARP levels in a dose-dependent manner. (**C**) UNC0379-treated cells showed prolonged sub-G1 phase and cell cycle arrest. (**D**) HEC50B and HEC1B cells were treated with 0.02% DMSO and UNC0379 for 9 days. UNC0379 treatment suppressed the growth of colonies (>100 cells) in a dose-dependent manner. (**E**) UNC0379-treated HEC50B and HEC1B cells were treated with either doxorubicin or cisplatin. Combinatorial treatment with UNC0379 and doxorubicin or cisplatin additively inhibited cell proliferation.

**Figure 3 cancers-14-05367-f003:**
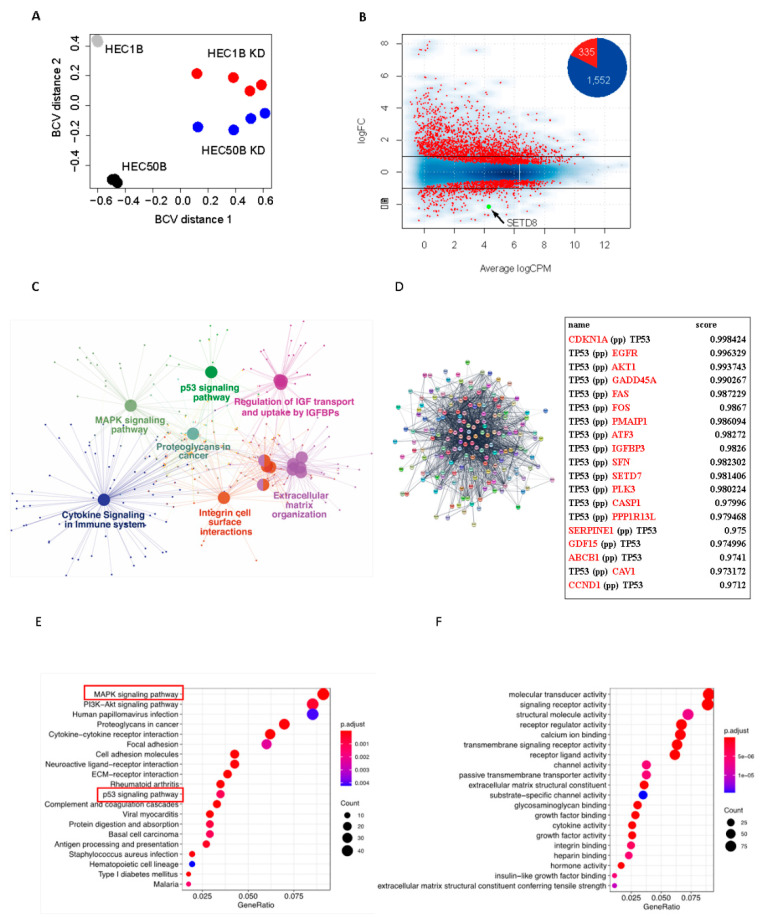
SETD8 regulates several genes from the p53 signal pathway. (**A**) Multidimensional scaling plot between individual RNA-seq. Distances between individuals correspond to the leading BCV. HEC1B and HEC50B cells treated with negative control siRNA (*n* = 3) are shown in gray and black, respectively. HEC1B and HEC50B cells treated with four SETD8-targeting siRNAs are shown in red and blue, respectively. (**B**) MA plots showing differentially expressed genes (DEGs: false discovery rate (FDR) < 0.05, red dots) between HEC50B cells treated with negative control (*n* = 3) and SETD8-targeting siRNA (*n* = 4). The differentially expressed *SETD8* is indicated by the arrow (green dot). Black lines indicate the log_2_ fold change (logFC) at 1 and −1. Average logCPM: the average log_2_ count per million. Circle plot shows the number of DEGs. (**C**) Relationships between enriched pathways. Pathway enrichment analysis for DEGs (FDR < 0.05, logFC > 1, *n* = 1552) in SETD8 knockdown HEC50B cells. For pathway enrichment analysis, Gene Ontology networks were generated using ClueGO and Cytoscape as described in the method section. (**D**) Protein-protein interactions (PPIs) of DEGs in SETD8-knockdown HEC50B cells. PPIs were visualized using Cytoscape with string database. First neighbors of TP53 are shown (FDR < 0.05, logFC > 1, *n* = 175). Top score of first neighbors of TP53 are shown. (**E**) KEGG pathway enrichment analysis for genes with upregulated expression (*n* = 1552) in SETD8-knockdown HEC50B cells. Dot plot showing GeneRatio (ratio of input genes that were annotated in a term) in the x-axis and terms sorted by GeneRatio in the y-axis. The adjusted *p*-value (*p*.adjust) is displayed as a gradient from red to blue. The number of counts is indicated by the size of the black circle. The common pathways identified in (**C**,**E**) are highlighted with red squares. (**F**) Gene Ontology analysis of molecular function for upregulated genes (*n* = 1552) in SETD8-knockdown HEC50B cells.

**Figure 4 cancers-14-05367-f004:**
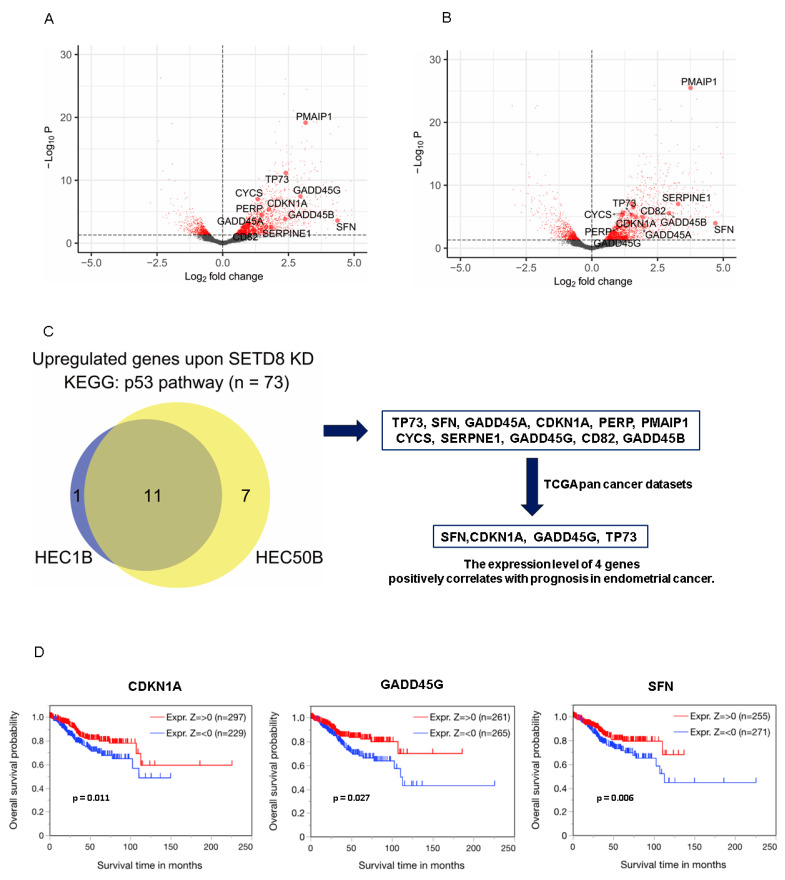
SETD8 regulates the expression of p53-related genes associated with endometrial cancer prognosis. (**A**) Volcano plots illustrating DEGs (FDR < 0.05, |logFC| > 1, red dots) between HEC1B cells treated with negative control (*n* = 3) and SETD8-targeting siRNA (*n* = 4). Differences in Log_2_ fold change in gene expression values are plotted on the x-axis. Adjusted *p*-values calculated using the Benjamin–Hochberg method are plotted on the y-axis. Genes corresponding to the KEGG p53 pathway are shown as large circles. (**B**) The same volcano plots as in (**A**) depicting the results from the HEC50B cell line. (**C**) Refinement of p53 gene regulated by SETD8. KEGG pathway analysis identified 11 genes associated with p53 that were significantly elevated in both HEC1B and HEC50B cell lines following SETD8 knockdown. Among the 11 genes, we identified four genes whose expression was positively correlated with the prognosis of endometrial cancer clinical specimens (TCGA). Overlapping *p*-value = 1.6 × 10^−35^. Overlap was tested using Fisher’s exact test. (**D**) Expression of *CDKN1A*, *SFN*, and *GADD45G* in the TCGA database and their influence of endometrial cancer prognosis.

**Figure 5 cancers-14-05367-f005:**
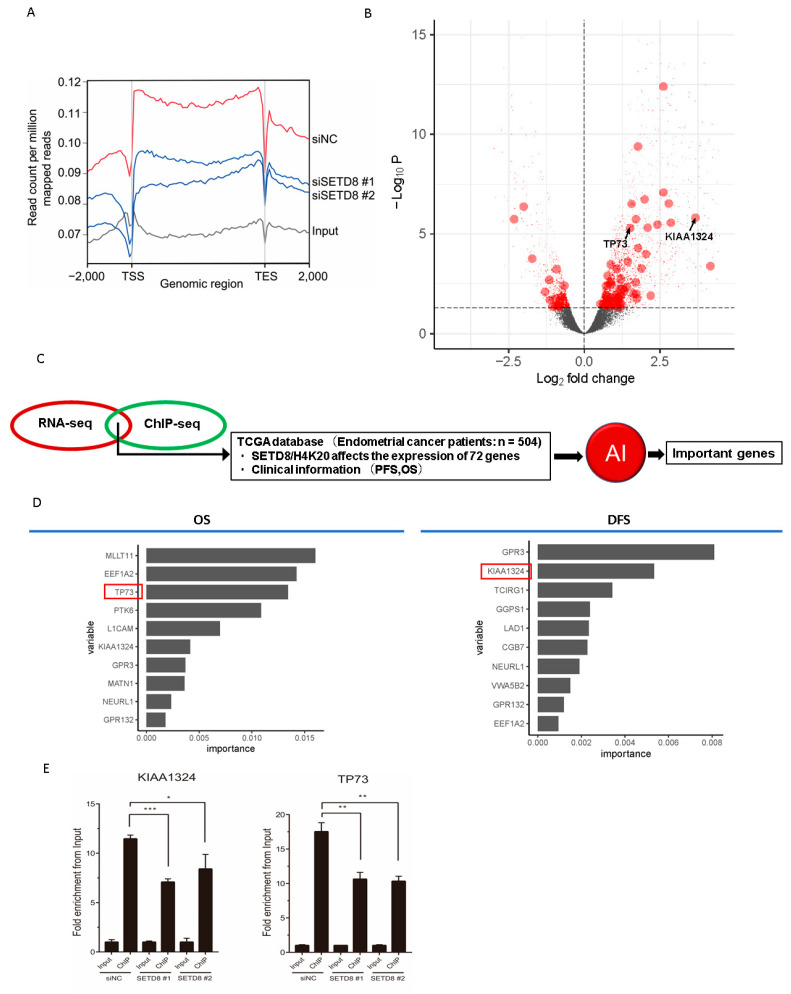
Machine learning approach to narrow down the critical genes regulated via SETD8/H4K20 methylation in endometrial cancer. (**A**) H4K20me1 ChIP-seq tag densities across different experiments. Density plot of the genomic region is shown as the read count per million mapped reads on the y-axis from 2000 bp upstream of the TSS to 2000 bp downstream of the transcription end site (TES) on the x-axis. The results for HEC50B cells treated with negative control siRNA (siNC, red line) or SETD8-targeting siRNA (#1 or #2, blue lines) are shown. Input is represented by the grey line. (**B**) Volcano plots illustrating DEGs (FDR < 0.05, red dots). H4K20me1 target genes annotated between 0–2 kb from TSS are shown as large circles. *TP73* and *KIAA1324* genes are indicated by the arrows. (**C**) Flowchart for narrowing down important genes using AI. RNA-seq and ChIP-seq results identified genes regulated via SETD8/H4K20 methylation. TCGA database was used to narrow down the genes related to OS and DFS prognosis of endometrial cancer using the random forest method. (**D**) The important genes regulated by SETD8 and related to the prognosis of endometrial cancer were identified using random forest method. (**E**) H4K20me1 occupancy at the *KIAA1324* or *TP73* gene promoter regions. HEC50B cells for ChIP-qPCR were treated with negative-control siRNA (NC) or SETD8-targeting siRNAs (SETD8 #1 and #2), respectively. Bar plots are shown as fold enrichment from Input (y-axis). * *p* < 0.05, ** *p* < 0.01, *** *p* < 0.001.

**Figure 6 cancers-14-05367-f006:**
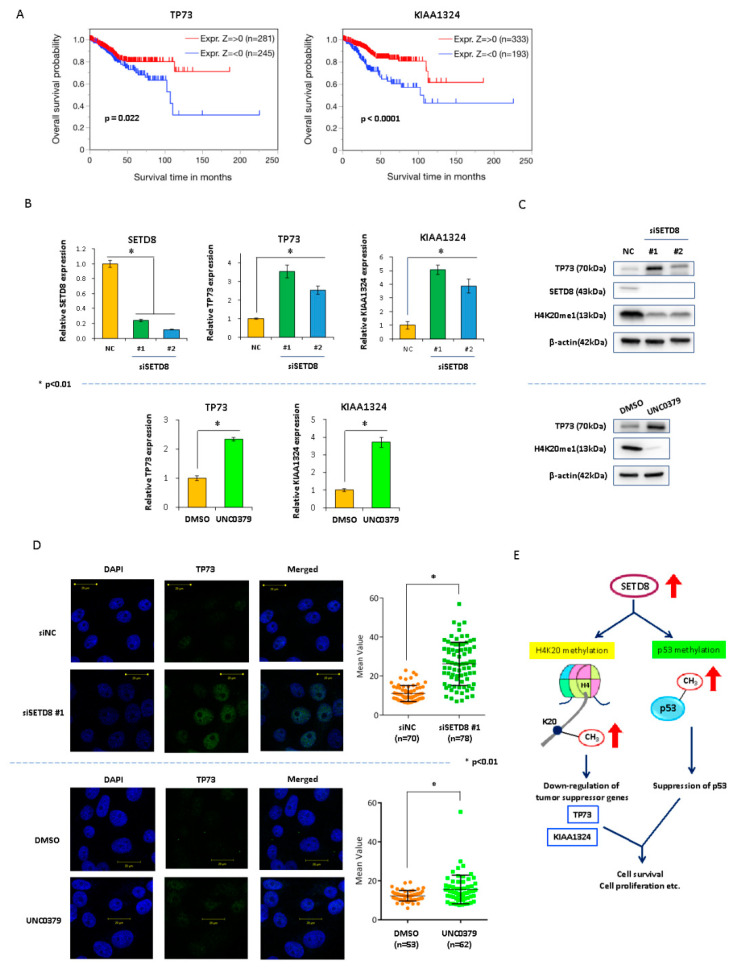
SETD8 regulates apoptosis-related genes, TP73 and KIAA1324 via H4K20 methylation in endometrial cancer cells. (**A**) Survival analysis of SETD8, TP73, and KIAA1324 in endometrial cancer using the TCGA datasets. (**B**) HEC50B cells were transfected with siSETD8 and treated with 0.02% DMSO as well as UNC0379 (3 μM) for 48 h. mRNA expression levels of *TP73* and *KIAA1324* were significantly upregulated in siSETD8-transfected and UNC0379-treated HEC50B cells (*p* < 0.01). (**C**,**D**) HEC50B cells were transfected with siSETD8 for 48 h and treated with 0.02% DMSO as well as UNC0379 for 96 h. SiSETD8-transfected and UNC0379 (1 μM)-treated cells showed a decrease in H4K20me1 and increase in TP73 protein expression levels. Following immunocytochemistry, siSETD8-transfected and UNC0379 (3 μM)-treated cells showed upregulated expression of *TP73*. (**E**) Schematic representation of SETD8-mediated regulation of *TP73*, *KIAA1324*, and *TP53* expression.

## Data Availability

The ChIP-seq and RNA-seq data utilized in this study have been deposited in the DNA Data Bank of Japan (DDBJ) with the accession number DRA013318.

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
