# Peer review of "The Histone Methyltransferase SETD8 Regulates the Expression of Tumor Suppressor Genes via H4K20 Methylation and the p53 Signaling Pathway in Endometrial Cancer Cells"

_cancers, 2022, doi:10.3390/cancers14215367_

Round 1

Reviewer 1 Report

        SETD8 regulates expression of multiple genes that are highly associated with the PS3 pathway is written very concise and completely. However, there is  lack of overall systematic expression. Since Fig. 3 is the most important issue in this article, It hoped that can be described in detail and integrated more completely!

        Fig. 5 shows the integration of 72 genes regulated by SETD8 via H4K20 methylation, using the TCGA database and combining with AI big data analysis, which is also the most important issue in this article, too. I hope that this problem can be further analyzed in detail to make the article more complete.

Author Response

We thank you and the reviewers for your thoughtful suggestions and insights. The manuscript has benefited from these insightful suggestions. I look forward to working with you and the reviewers to move this manuscript closer to publication in Cancers.

The manuscript has been rechecked and the necessary changes have been made in accordance with the editor and reviewer’s suggestions. The responses to all comments have been prepared as below, and all changes in the revised manuscript are highlighted in yellow.

Thank you for your consideration. We look forward to hearing from you.

1.SETD8 regulates expression of multiple genes that are highly associated with the PS3 pathway is written very concise and completely. However, there is lack of overall systematic expression. Since Fig. 3 is the most important issue in this article, It hoped that can be described in detail and integrated more completely!

Response: Thank you very much for your accurate remarks. We have added the following content in the text to systematically express the overall fluctuation in gene expression. In addition, the common pathways obtained in Figures 3C and E are marked using red squares.

Lines 348-349: Notably, we found that these pathways are not only significantly enriched but also systematically connected (Figures 3C and E).

Lines 354-355: consistent with the systematic interconnection among the pathways we identified above (Figures 3C and E).

2.Fig. 5 shows the integration of 72 genes regulated by SETD8 via H4K20 methylation, using the TCGA database and combining with AI big data analysis, which is also the most important issue in this article, too. I hope that this problem can be further analyzed in detail to make the article more complete.

Response: Thank you for this valuable comment. In this case, machine learning was used only as an experimental method used to select important genes. The purpose of using this method was not to create a risk prediction model. However, the risk assessment itself using machine learning is useful, and we plan to work on this aspect in future research.

We have revise manuscript according to your comment (Lines 546-549)

Reviewer 2 Report

-Journal -  Cancers (ISSN 2072-6694)
cancers-1962511
-Type: article
-Title: Prognostic genes for endometrial cancer identified using machine learning on gene clusters regulated by SETD8 histone methyltransferase  

In this work, Kukita and colleagues investigated the implication of the histone methyltransferase SET domain-containing protein 8 (SETD8) inn endometrial cancer development in vitro. The authors (i) evaluated the expression profile of SETD8 in cells and (ii) evaluated functionally, by small interfering RNAs the role of this gene in regulating endometrial cell proliferation and apoptosis. Mai results indicate that SETD8 regulates genes via H4K20 methylation and the p53 signaling pathway. In addition, by using a machine learning approach, KIAA1324 and TP73 genese have also identified as prognostically important genes related to apoptosis in endometrial cancer. This is a very well written and organized work, reporting novel important findings on the epigenetics of endometrial cancer, which is an important topic considering the therapeutic and prognostic potential of the reported findings. I have several minor observations.

 Here some minor observations

First introductive lines. For a better reading, I suggest including the currently know 5 years survival rate of endometrial cancer and a couple of words on its epidemiology.
Lines 64-66 Histone modifications have bene reported to play and important role in epithelial tumors. For completeness this additional supporting reference should be included https://doi.org/10.3390/ijms222111464
Lines 69-70 concerning the EZH2 inhibitors, DZNep should be mentioned as a potent antitumor agent
Line113 the authors should justify the case/control imbalance as only n=4 endometrium samples have bene included in the analysis. It is difficult to observe the statistically significant difference reported in figure 1a
Methods should be shortened by at least 20%, while unnecessary information can be moved to supplemental. Moreover, methods are completely lacking in supporting references. They should be included in the majority of methods sections, including statistics, from 2.3 to 2.17.
Line 407 I am not sure whether RSF is Random forest survival (RSF),  in any case its complete name should be included, if available
421 instead of “Use of machine learning to narrow” I would say “Machine learning approach to narrow”
Line 465 SETD8 also play a role in ovarian cancer doi: 10.3390/biom10121686 . Its considered a potential novel therapeutic target for this gynecological tumor

Author Response

We thank you and the reviewers for your thoughtful suggestions and insights. The manuscript has benefited from these insightful suggestions. I look forward to working with you and the reviewers to move this manuscript closer to publication in Cancers.

The manuscript has been rechecked and the necessary changes have been made in accordance with the editor and reviewer’s suggestions. The responses to all comments have been prepared as below, and all changes in the revised manuscript are highlighted in yellow.

Thank you for your consideration. We look forward to hearing from you.

In this work, Kukita and colleagues investigated the implication of the histone methyltransferase SET domain-containing protein 8 (SETD8) inn endometrial cancer development in vitro. The authors (i) evaluated the expression profile of SETD8 in cells and (ii) evaluated functionally, by small interfering RNAs the role of this gene in regulating endometrial cell proliferation and apoptosis. Mai results indicate that SETD8 regulates genes via H4K20 methylation and the p53 signaling pathway. In addition, by using a machine learning approach, KIAA1324 and TP73 genese have also identified as prognostically important genes related to apoptosis in endometrial cancer. This is a very well written and organized work, reporting novel important findings on the epigenetics of endometrial cancer, which is an important topic considering the therapeutic and prognostic potential of the reported findings. I have several minor observations.

 Here some minor observations

1.First introductive lines. For a better reading, I suggest including the currently know 5 years survival rate of endometrial cancer and a couple of words on its epidemiology.

Response: We appreciate your important suggestion. We have revised the manuscript according to your advice (Lines 57-59)

2.Lines 64-66 Histone modifications have bene reported to play and important role in epithelial tumors. For completeness this additional supporting reference should be included https://doi.org/10.3390/ijms222111464

Response: We appreciate this important suggestion. We have revised the manuscript according to your advice (Lines 65-66).

3.Lines 69-70 concerning the EZH2 inhibitors, DZNep should be mentioned as a potent antitumor agent.

Response: We appreciate this important suggestion. We have revised the manuscript according to your advice(Lines 73-74).

4.Line113 the authors should justify the case/control imbalance as only n=4 endometrium samples have bene included in the analysis. It is difficult to observe the statistically significant difference reported in figure 1

Response: We appreciate this valuable suggestion. The word "significantly" was not appropriate given the low number of controls; we have revised the manuscript accordingly. The reason for the low number of controls is the difficulty in obtaining normal endometrial samples from patients(Lines 263 and Lines 442)

5.Methods should be shortened by at least 20%, while unnecessary information can be moved to supplemental. Moreover, methods are completely lacking in supporting references. They should be included in the majority of methods sections, including statistics, from 2.3 to 2.17.

Response: In accordance with your suggestion, we have shortened the Methods section accordingly and added references. In addition, the section "2.17. Statistical Analysis" has been revised to clarify the methods used for statistical analyses.

6.Line 407 I am not sure whether RSF is Random forest survival (RSF), in any case its complete name should be included, if available

Response: We appreciate this important suggestion. We have revised the manuscript according to your advice (Lines380)

  1. 421 instead of “Use of machine learning to narrow” I would say “Machine learning approach to narrow”

Response: We appreciate this important suggestion. We have revised the manuscript according to your advice (Lines 395).

Line 465 SETD8 also play a role in ovarian cancer doi: 10.3390/biom10121686 . Its considered a potential novel therapeutic target for this gynecological tumor

Response: We appreciate this important suggestion. We have revised the manuscript according to your advice (Lines 439).

Reviewer 3 Report

Research presented in the manuscript investigates the role of SETD8 and its regulation of other genes in endometrial cancer.

Major comments:

1.     Manuscript title includes “machine learning” but machine learning is one of the techniques along with other tools and techniques used in this study, therefore the title does not justify the manuscript.

2.     The introduction section provides very little to no information on how and which machine learning was used. It is recommended to provide more relevant information on machine learning and how it was used in the research.

3.     It is not clear which ML model was used? What is the sensitivity, specificity, AUROC of the ML prediction model used in this manuscript? How the model used in this manuscript is better than other ML models?

It is highly recommended that authors submit more details on the ML applied in this study so that reviewers and readers can replicate the study, if required.

Author Response

We thank you and the reviewers for your thoughtful suggestions and insights. The manuscript has benefited from these insightful suggestions. I look forward to working with you and the reviewers to move this manuscript closer to publication in Cancers.

The manuscript has been rechecked and the necessary changes have been made in accordance with the editor and reviewer’s suggestions. The responses to all comments have been prepared as below, and all changes in the revised manuscript are highlighted in yellow.

Thank you for your consideration. We look forward to hearing from you.

Research presented in the manuscript investigates the role of SETD8 and its regulation of other genes in endometrial cancer.

Major comments:

  1. Manuscript title includes “machine learning” but machine learning is one of the techniques along with other tools and techniques used in this study, therefore the title does not justify the manuscript.

Response: We appreciate this important suggestion. We have revised the title accordingly; it now reads as follows: “The histone methyltransferase SETD8 regulates the expression of tumor suppressor genes via H4K20 methylation and the p53 signaling pathway in endometrial cancer cells.”

  1. The introduction section provides very little to no information on how and which machine learning was used. It is recommended to provide more relevant information on machine learning and how it was used in the research.

Response: We appreciate this important suggestion. We have revised the manuscript according to your advice. We have provided more relevant information regarding machine learning and how it was used in the research (Lines104-110 and Lines115-117)

  1. It is not clear which ML model was used? What is the sensitivity, specificity, AUROC of the ML prediction model used in this manuscript? How the model used in this manuscript is better than other ML models? It is highly recommended that authors submit more details on the ML applied in this study so that reviewers and readers can replicate the study, if required.

Response: Thank you for your advice. We added the detailed description of these methods according to your advice (Lines239-242). Random survival forest is a machine-learning algorithm that calculates the hazard function as an ensemble of hazard functions estimated by survival trees. Like the random forest method, RSF is robust to outliers and can accurately assess the risk of event occurrence based on multiple factors. Furthermore, because random survival forest is a survival time analysis, sensitivity, and specificity cannot be calculated, but the c-index corresponding to the AUROC can be calculated. Accordingly, we analyzed the c-index (Lines385-387)

Round 2

Reviewer 3 Report

The authors tried to incorporate the changes in the revised manuscript. Specifically, the limitation of the RSF model should be considered carefully in such analysis as it needs a large number of features for accurate prediction.

Author Response

We thank you and the reviewers for your thoughtful suggestions and insights. The manuscript has benefited from these insightful suggestions. I look forward to working with you and the reviewers to move this manuscript closer to publication in Cancers.

The manuscript has been rechecked and the necessary changes have been made in accordance with the editor and reviewer’s suggestions. The responses to all comments have been prepared as below, and all changes in the revised manuscript are highlighted in green.

Thank you for your consideration. We look forward to hearing from you.

Editor(s)' Comments to Author:

Reviewer3

The authors tried to incorporate the changes in the revised manuscript. Specifically, the limitation of the RSF model should be considered carefully in such analysis as it needs a large number of features for accurate prediction.

Response 3

Response: We appreciate this important suggestion. We have revised the manuscript according to your advice (Lines549-553).